# Rethinking 2D segmentation network to segment CT scans

Heonjin Ha
math.hjha@gmail.com

**Abstract.** This work experiments with how 2D segmentation models work to segment CT scans. To overcome the lack of spatial information, the stacking-slices method is devised. This performs similar to 3D segmentation models like 3D-UNet. Moreover, this work suggests a new post-processing technique, and this can keep the organs in the background from being removed.

**Keywords:** 2D segmentation, kits21 dataset

## 1  Introduction

Abdominal CT scans consists of 3D voxels. And organs and components of the human body are also 3D objects. If I see them with 2D slices, then I lose spatial information. Nevertheless, I approached this segmentation problem with 2D models. Rapid progress has been made in 2D image segmentation. Various resolution aggregation methods[1, 2] are able to detect small and big 2D objects. Transformer networks[3] are also suggested for solving segmentation problems. Furthermore, 2D segmentation models are more memory-efficient than 3D ones. With these benefits, I achieved better performance than plain 3D UNet models. To obtain the 3D spatial information, I optimized our stacking-slices strategy.

By looking at the given KITS21 dataset, the two classes, tumor and cyst were found to occupy a part of the kidney. So, I used three different models on three classes and combined them into one 3D output. This methodology is useful to keep the organs that can be removed by post-processing.

## 2  Methods

### 2.1  Training and Validation Data

My submission made use of the official KiTS21 training set alone. But I'd like to use other datasets in the future.

### 2.2  Preprocessing

First of all, I clipped HU values with above 400 and below -400 and then did mini-max normalization. I chose these values by heuristically seeing the CT scans with various threshold values.  I utilized 2D augmentation methods. For example, I added multiplicative noise, random brightness contrast, horizontal Flip, and Shift Scale Rotation. I only trained on slices where each slice has a class segmentation mask, not a background mask only.

### 2.3  Proposed Method

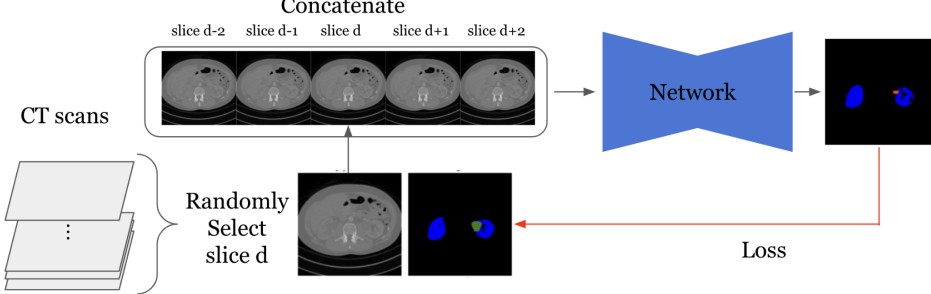

Figure 1. Training process.

Here, I see axial views of CT scans. I randomly chose 5 successive slices and made models predict the mask of their middle slice. To be specific, let's say D is the number of slices in a CT scans, and then choose $d$ such that $0 \leq d \leq D$. This slice $d$ has a non trivial mask, meaning it has classes more than one. I stack five successive slices $d-2, d-1, d, d+1$, and $d+2$ and these stacked slices are the model's input.

Feature Pyramid Network[4] with EfficientNetB3[5] is my backbone, and I will further use other Network architectures. Cross-Entropy Loss is used as a penalty function of kidney and tumor classes. I used Combo Loss[6] to Cyst which is a summation of Cross-Entropy Loss and Dice Loss. Adam Optimization is used but will be changed. The maximum learning rate is 7e-4 and the initial learning rate of OneCycleLR is one-tenth of it and the final learning rate is one-thirtieth of it. I trained 1700 epochs for each model. I run 5-fold cross-validation, and I chose the model of the best Dice score for each class. I trained three models on three classes and aggregated them. I will run more models to ensemble them which I don't decide how. For cyst, because it is hard to detect, I gathered all the segmented pieces of the cysts from the different 5-fold cross-validation models.

## 3  Results

Metric values during validation I see are IoU, Dice score, pixel accuracy on each class. Moreover, to keep following my training works well, I track the gradient of each weight and bias, validation loss.
The following gifs are not from the final model but just interim results of a 2D segmentation model of CT scans. You can see there is redundant kidney/tumor segmentation, but post-processing will delete them out.

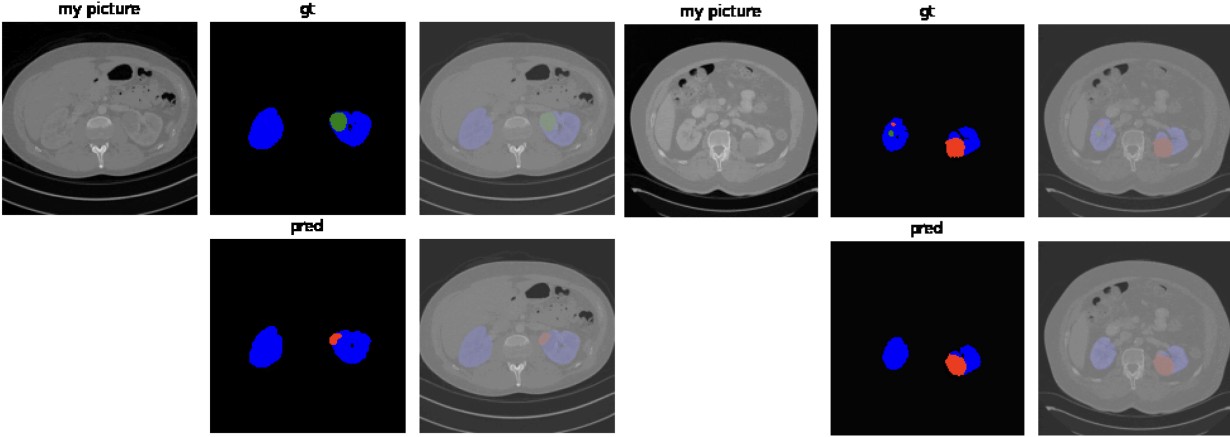

Figure 2. Qualitative result.

I tuned hyperparameters on each class, such as the number of iterations, and warm-up schedulers. Table 1. is the validation Dice result on one of 5-fold cross-validations. This result is not a post-processed one.

|  |  | Kidney | Tumor | Cyst |
|---|---|---|---|---|
| tuned class | Kidney | 0.7265 | 0.4693 | 0.0154 |
|  | Tumor | 0.6958 | 0.5284 | 0.0424 |
|  | Cyst | 0.709 | 0.2543 | 0.0824 |

Table 1. Quantitative result.

## 3  Discussion and Conclusion

Cysts are observed in more than one connected component, so it is careful to do the post-processing. For me, the

volume of the cyst is small so that using 2D segmentation is powerful in this case. 2D segmentation models to solve 3D objects need to be further analyzed. Finding patterns with 2D segmentation models is enough in the sense that the models already have a lot of trainable parameters.

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
