# OpenReview forum: "Rethinking 2D segmentation network to segment CT scans"
_MICCAI.org/2021/Challenge/KiTS — Submitted to KiTS21 Challenge_

### Official Review · Reviewer_yHg7 · 2021-08-30

**Rating:** 6

**Review:**

The authors present an approach that utilizes a 2d model to segment the 3d volumes by simply predicting based on a stack of 5 consecutive frames of the 3d object. By doing this, the authors have access to a large collection of pre-trained 2d architectures that they can use for initialization. Also, the smaller memory footprint could potentially allow them to free up more resources for experimentation and training which could ultimately help them to outperform 3d models.

Personally, I favor 3d approaches for data that is inherently 3d, but I respect the author's unconventional approach and the arguments they make for its potential benefits. At the time the authors wrote this, they did not seem to have detailed results available. They should be sure to include these in their revised paper. Also, other sections are relatively short. The authors should aim to expand on their arguments generally. They should consult the official paper template for a list of topics that they should address. Many of these are currently omitted.

---

### Official Review · Reviewer_yPf6 · 2021-08-30

**Rating:** 7

**Review:**

### Overall

- An institutional email address is preferred over gmail if possible
- You switch between saying "I" and "we". Please be consistent

### Introduction

- Looks good

### Methods

- When you say you clipped values at 400, did you only clip above? Or did you use a bound below as well? Why did you choose this/these values?
- Did you account for different slice thicknesses or other spacing at all with e.g. resampling?
- It would be nice if you could include a figure that shows a diagram of your model, especially as it pertains to using a stack of 2d slices as input to a 2d network

### Results

- It's very nice that you have examples of predictions compared to their corresponding ground truth, but it would be great if you could also provide a table with your validation results and also, once known, add your official test set performance

---

### Decision · Program_Chairs · 2021-08-30

**Decision:**

Major Revisions

**Comment:**

Please address the reviewer comments and resubmit